# Persistent Southern Tomato Virus (STV) Interacts with Cucumber Mosaic and/or Pepino Mosaic Virus in Mixed- Infections Modifying Plant Symptoms, Viral Titer and Small RNA Accumulation

**DOI:** 10.3390/microorganisms9040689

**Published:** 2021-03-26

**Authors:** Laura Elvira González, Rosa Peiró, Luis Rubio, Luis Galipienso

**Affiliations:** 1Biotechnology and Plant Protection Center, Valencian Institute of Agricultural Research (IVIA), 46113 Valencia, Spain; elviragonzalez.laura@gmail.com (L.E.G.); lrubio@ivia.es (L.R.); 2Biotechnology Department, Universitat Politècnica de València, 46022 Valencia, Spain; ropeibar@btc.upv.es

**Keywords:** persistent virus, *Amalgaviridae*, synergism, antagonism, vsiRNAs, miRNAs, mixed-infections

## Abstract

Southern tomato virus (STV) is a persistent virus that was, at the beginning, associated with some tomato fruit disorders. Subsequent studies showed that the virus did not induce apparent symptoms in single infections. Accordingly, the reported symptoms could be induced by the interaction of STV with other viruses, which frequently infect tomato. Here, we studied the effect of STV in co- and triple-infections with Cucumber mosaic virus (CMV) and Pepino mosaic virus (PepMV). Our results showed complex interactions among these viruses. Co-infections leaded to a synergism between STV and CMV or PepMV: STV increased CMV titer and plant symptoms at early infection stages, whereas PepMV only exacerbated the plant symptoms. CMV and PepMV co-infection showed an antagonistic interaction with a strong decrease of CMV titer and a modification of the plant symptoms with respect to the single infections. However, the presence of STV in a triple-infection abolished this antagonism, restoring the CMV titer and plant symptoms. The siRNAs analysis showed a total of 78 miRNAs, with 47 corresponding to novel miRNAs in tomato, which were expressed differentially in the plants that were infected with these viruses with respect to the control mock-inoculated plants. These miRNAs were involved in the regulation of important functions and their number and expression level varied, depending on the virus combination. The number of vsiRNAs in STV single-infected tomato plants was very small, but STV vsiRNAs increased with the presence of CMV and PepMV. Additionally, the rates of CMV and PepMV vsiRNAs varied depending on the virus combination. The frequencies of vsiRNAs in the viral genomes were not uniform, but they were not influenced by other viruses.

## 1. Introduction

Southern tomato, Pepino mosaic, and Cucumber mosaic viruses (STV, PepMV, and CMV, respectively) infect tomato *(Solanum lycopersicum)* crops worldwide. CMV and PepMV are two pathogenic or acute viruses that are responsible for important economic losses [1,2]. STV is a persistent double-stranded RNA (dsRNA) virus belonging to the genus *Amalgavirus* (family *Amalgaviridae*), whose genome is 3.5 kb in length, which contains two overlapping open reading frames (ORFs): ORF 1 encodes for the 42 kDa putative coat protein (CP or p42) and ORF 2 encodes for the RNA-dependent RNA-polymerase (RdRp) by +1 ribosomal frameshifting [3,4]. STV is only transmitted by seed, with rates up to 80% and no viral particles have been detected until now [3,5,6]. High virus incidence has recently been reported in two important Spanish tomato producer areas, such as the Gran Canarias and Valencian Community [5,7]. Despite that STV was first associated with some fruit symptoms, such as lack of maturation and color alterations, it was recently shown that the virus is not responsible of any apparent plant symptom in tomato plants infect by only STV [5,6]. Hence, the reported symptoms could be induced by other pathogenic or acute viruses, such as PepMV, CMV, or Tomato mosaic virus (ToMV), which frequently appear in mixed infection with STV in tomato crops or by interaction of STV with other viruses [8,9].

PepMV is a (+) polarity single stranded RNA (ssRNA+) virus belonging to the genus *Potexvirus* (family *Flexiviridae*) whose genome is 6.4 kb in length and contains five ORFs: ORF 1 encodes for the RdRp, ORF 2, 3, and 4 for the triple gene block proteins (TGB), involved in virus movement, and ORF 5 for the CP [10,11,12]. PepMV induces symptoms of leaf mosaic and alteration of fruit color and maturation, but the symptom severity depends on several factors, such as the virus strain and crop conditions. PepMV is transmitted by contact and by seed with very low rates up to 0.06% [13,14]. No commercial tomato varieties with natural resistances against PepMV are available, so disease control has only been achieved by cross protection with mild PepMV strains [15,16].

CMV is a tripartite ssRNA+ virus that belongs to the genus *Cucumovirus* (family *Bromoviridae*): RNA 1 is 3.4 kb in length that contains the ORF 1a encoding a RdRp subunit; RNA 2 is 3.1 kb in length and contains the overlapping ORFs 2a y 2b, encoding the other RdRp subunit and the RNA silencing suppressor (VSR) 2b protein; RNA 3 is 2.2 kb in length and it contains the two separated ORFs 3a and 3b encoding for the cell-to-cell movement protein and CP, respectively [17]. CMV infects a broad spectrum of plants species (more than 1200 plant species in 100 families), including tomato and pepper *(Capsicum annuum)* and the main way of virus transmission is by aphids in a semi-persistent manner. Symptoms that are induced by CMV depend on the host species and the presence of RNA satellite molecules: in tomato, the most common symptoms induced by CMV are plant stunting, mosaic, and leaf deformation, but the presence of the CARNA-5 satellite enhances the disease severity, inducing leaf and fruit necrosis and plant death [18,19]. There are no commercial tomato varieties with natural resistances against CMV, and the only manner to minimize the CMV impact is by controlling the aphid populations into the crops.

When plants are infected by RNA viruses, viral dsRNAs (generated during virus replication) activate the post transcriptional gene silencing (PTGS), a plant defense mechanism that produces the degradation of invasive RNAs in small molecules of 21–24 nt (Virus small interfering RNAs, vsiRNAs). PTGS is also involved in the degradation of highly structured plant mRNA rendering micro RNAs (miRNAs), which are small RNA molecules that are equivalent in length to vsiRNAs. miRNAs are involved in the regulation of gene expression in many crucial plant processes, such as development, reproduction, and stress. The modification of the miRNA expression level could lead to disease development [20,21]. In addition, vsiRNAs that are derived from viruses could mimic plant miRNAs by sequence homology targeting and regulating post-transcriptionally some host genes [22,23]. In the case of persistent viruses, the information about the effect of viral infection regarding on both vsiRNA and miRNAs populations is scarce. The low production of vsiRNAs in plants infected with STV has recently been reported, but the virus can modify the populations of some miRNAs in tomato plants [6].

Mixed-infections with two or more plant viruses are frequent in fields and they can interact in multiple and complicate ways [24]. The interaction can be synergistic, increasing the replication of at least one of the viruses and/or enhancing symptoms. Synergistic interactions are known to be predominantly produced by unrelated viruses that infect the same host cells. The mechanism underlying the synergistic relationships are not well determined, but numerous viral and/or host products might be involved. The best characterized are those involving potyviruses (genus *Potyvirus*, family *Potyviridae*) as one of the viral partners. In this case, potyviral VSRs are involved in the increase of multiplication and plant symptom enhancing of other viral partner [25,26]. In the opposite site, antagonistic interactions between closely viruses (cross protection or mutual exclusion) may occur. In the cross protection, a previous infection with one (protecting) virus prevents or interferes with the subsequent infection by other homologous (challenging) virus [16,27] whereas, in the mutual exclusion, two or more viruses infect simultaneously a plant. Several mechanisms have been proposed for the cross protection phenomenon, such as the CP of the protecting virus can prevent the CP disassembly of the challenging virus, which is necessary for infection or the sequence-specific degradation of the challenging virus RNA as consequence of PTGS activation by the protecting virus [28,29]. The mechanism for mutual exclusion is still obscure, but it has been proposed that a plant might be considered to be an environment structured spatially for plant virus infections, and cells could only become infected by only one virus [30].

The number of studies on viruses in mixed infection has increased lately, providing valuable knowledge that may be useful in controlling complex diseases. However, information regarding interactions between persistent and acute viruses in is very scarce. In this work, we studied the interactions of the persistent STV and the acute PepMV and CMV in tomato. Plant symptoms, virus RNA accumulation, and miRNA and vsiRNA accumulation were assessed in single, double, and triple infections.

## 2. Materials and Methods

### 2.1. Plant Material, Virus Infection Assay and Sample Preparation

Tomato seedlings var. Roque were analyzed by RT-qPCR to determinate the presence and/or absence of STV [5]. The absence of ToMV and PepMV, the main tomato seed-borne viruses, was assessed by conventional RT-PCR and RT-qPCR, respectively [31,32]. PepMV and CMV isolates (kindly provided by Drs. A. Alfaro and M.I. Font) were collected in tomato fields from Southern Spain in 2015 and 2016, respectively, and maintained in *Nicotiana benthamiana* plants. To exclude possible mixed infections with other viruses, the plants that were infected with these CMV and PepMV isolates were tested by ELISA for the most common viruses infecting tomato in the collection region, such as CMV, ToMV, Tomato spotted wilt virus (TSWV), and Parietaria mottle virus (PMoV) [33,34,35,36,37].

Mechanical inoculation was performed by the homogenization of 1 g of CMV or PepMV infected *N. benthamiana* plants in inoculation buffer (0.01M Na_2_HPO_2_ and 0.01M Na_2_HPO_4_, pH 7.2) and rub-inoculation by using carborundum in the two first tomato (var. Roque) true leaf [38]. For double infection with CMV and PepMV, the tomato plants were mechanically inoculated with an equivalent mix (*w*/*w*) of *N. benthamiana* plants infected with each virus. The assay consisted of a total of 71 tomato plants with the following virus combinations: five, eight, and 10 plants were single-infected with PepMV, CMV, and STV, respectively; 10 plants were co-infected with STV and PepMV, 10 were co-infected with STV and CMV, and eight were co-infected with CMV and PepMV, and, finally, 10 plants were triple-infected with STV, CMV, and PepMV. As control, 10 plants were mock-inoculated by using only the inoculation buffer. Because STV is not a mechanically transmitted virus, tomato plants that tested positive for STV by RT-qPCR were used as STV-single infected plants or were inoculated with CMV and/or PepMV to obtain the corresponding co- or triple-infections. Tomato plants were kept in a greenhouse with ventilation and the presence and accumulation of STV, CMV, and PepMV was evaluated by RT-qPCR at five, 10, 15, and 20 days post inoculation (dpi). Plant symptoms consisting in leaf deformation and mosaic were recorded in this period. A scale of symptom severity was established scoring from 0 to 3, where 0 corresponded to no symptomless, and 1, 2, and 3 to mild, moderate, and severe symptoms, respectively (Figure 1). Plant height and weight were measured at the end of the experiment (20 dpi).

For sample preparation, 0.1 g of apical leaves were ground in a power homogenizer TissueLyser (Qiagen, Germany) with liquid nitrogen. The total RNA was extracted by using a phenol:chloroform:isoamyl alcohol standard protocol followed by ethanol precipitation [39].

### 2.2. Conventional RT-PCR and RT-qPCR Assays

CMV and PepMV conventional RT-PCR was performed from the total RNA extracts. The RNA extracts were denatured in the presence of 0.8 µM of the corresponding reverse primer and cDNA was obtained with the SuperScript IV kit (ThermoFisher, Waltham, MA, USA) at 55 °C for 20 min. and 80 °C for 10 min. PCR was done with 0.5 µM of the corresponding forward and reverse for each virus and the Taq polymerase kit (ThermoFisher, Waltham, MA, USA) following the manufacturer’s instructions. The PCR conditions were cDNA denaturation at 94 °C for 5 min., 35 cycles of DNA amplification at 94 °C for 30 s, 55 °C for 30 s, and 72 °C for 40 s, and a final DNA chain extension of 72 °C for 5 min. The amplification products were separated by electrophoresis in 2% agarose gels and visualized by UV after staining with GelRed (Sigma-Aldrich, San Luis, MS, USA). Specific PCR products were purified with Qiagen minElute PCR purification kit (Qiagen, Hilden, Germany) and sequenced by Sanger with an ABI 3130 XL capillary sequencer (Applied Biosystems, Foster City, CA, USA). CMV and PepMV nucleotide sequences were deposited in GenBank under the accession numbers MT785769 and MT785770, respectively.

STV quantification was performed by RT-qPCR with primers and TaqMan probe set previously designed in the CP (1189–1257 nts) region [5]. PepMV quantification was done using a primers and TaqMan probe set that was designed in a TGB2 conserved region (5126–5213 nts) that allowed for amplifying all virus isolates [32]. For CMV quantification, primers and TaqMan probe were designed by using the software Primer Express (ThermoFisher, USA) on basis of the CP nucleotide sequence (1533–1610 nts) that was obtained from conventional RT-PCR. RT-qPCR was performed with the One step PrimeScript RT-PCR Kit (TaKaRa, Shiga, Japan) in LightCyler 480 (Roche, Basilea, Switzerland) following the manufacturer instructions with some modifications. The total RNAs extracts (50 ng) were denaturalized in presence of 0.2 µM of both forward and reverse primers 95 °C for 5 min. Subsequently, a mix containing the 10 µL one-step RT-PCR buffer III, 2 U Ex Taq HS, 0.4 µL PrimeScript RT Enzyme Mix II, and 0.2 µM specific TaqMan probe was added to a final volume of 20 μL. The thermal cycling conditions were: reverse transcription at 42 °C for 15 min., incubation at 94 °C for 10 s, and 40 cycles of DNA amplification at 94 °C for 5 s and 60 °C for 20 s. The total RNA extracts of mock-inoculated tomato plants were used as negative RT-qPCR control. The specificity of all virus primer and probe sets were assessed to avoid unspecific cross-amplifications.

Appendix A shows all the primers and probe sequences and their respective applications.

### 2.3. Preparation of RNA Transcripts and Standard Curve

The templates for in vitro transcription were obtained by conventional RT-PCR from total nucleic acid extracts of STV-infected tomato and CMV- or PepMV-infected *N. benthamiana* plants, as described in Section 2.2. A modified version of the reverse primers (the T7 promoter sequence was added at the 5’-terminus) used for RT-qPCR were used for in vitro transcription (Appendix A). The transcription reaction was done with the Megascript T7 Kit (ThermoFisher, Waltham, MA, USA) following the manufacturer’s instructions. To eliminate the contaminant cDNA, the RNA transcript reaction was treated twice with RNasa free DNasa set (ThermoFisher, Waltham, MA, USA) and then purified by the phenol:chloroform:isoamyl method [39]. The final transcript concentration was estimated with a nanodrop 1000 spectrophotometer (ThermoFisher, Waltham, MA, USA), and molarity was assessed with the formula: pmol of ssRNA = μg of ssRNA × (106 pg/1 μg) × (1 pmol/340 pg) × (1/Nb), in which 340 is the average molecular weight of a ribonucleotide and Nb the number of bases of the transcript. The Avogadro’s constant (6.023 × 10^23^ molecules/mol) was used to calculate the number of RNA transcript copies. In order to generate external standard curves, 10-fold serial dilutions containing 10^11^–10^1^ RNA copies of each transcript in total RNA extracts from mock-inoculated tomato plants were analyzed by RT-qPCR. For each dilution, three repeats (technical replicates) were done, and the Ct mean value was calculated. Quantitative optimal range were obtained from 10^11^ to 10^4^ virus RNA copies/ng of total RNA for STV, from 10^10^ to 10^4^ virus RNA copies/ng of total RNA for CMV and from 10^11^ to 10^3^ copies/ng of total RNA for PepMV. For all of the viruses, standard curves showed a strong linear relationship with very high correlation coefficients of *R*^2^ = 0.99, low variation coefficient (<0.5%), and high amplification efficiencies (>99%).

### 2.4. High-Throughput Small RNA Sequencing

For the elaboration of the small RNA libraries, three independent biological replicates were used from tomato plants that were infected with STV, CMV, or PepMV, or the different virus combinations. Each biological replicate consisted of a mix of total RNA extracts that were obtained from two or three tomato plants at 15 dpi. As control, small RNA libraries from mock-inoculated plants were synthetized. RNA concentration and purity were determined using the Qubit^®^ RNA assay Kit in a Qubit^®^ 3.0 Fluorometer (ThermoFisher, Waltham, MA, USA) and the NanoPhotometer^®^ spectrophotometer (IMPLEN, Los Angeles, CA, USA), respectively. The RNA integrity was determined in the Agilent Bioanalyzer 2100 system with the RNA Nano 6000 assay Kit (Agilent Technologies, Santa Clara, CA, USA). cDNA was obtained from 1 µg of total RNA of each biological replicate by using the NEBNext^®^ Multiplex Small RNA library Prep Set for Illumina^®^ (Sigma Aldrich, San Luis, MS, USA) and then sequenced by using the Illumina NextSeq550 platform (Illumina, San Diego, CA, USA). cDNA libraries were uploaded to the NCBI platform and published under the Bioproject PRJNA625104 and PRJNA574043. The reads were cleaned by trimming the sequencing adapters and low-quality reads were filtered using SeqTrimNext software applying the standard parameters for Illumina short reads [40]. The biological replicate distribution was analyzed by Principal Component Analysis (PCA) to reduce the dimensionality of the dataset. The length of the reads was restricted from 21 to 24 nts. The identification and quantitation of miRNAs were performed through Oasis 2.0 pipeline analysis (https://tools4mirs.org/software/precursor_prediction/oasis/, Access date, 7 March 2021): reads were aligned with the STAR program in the database RNAbase 2.1 (ftp://mirbase.org/pub/mirbase/ Access date, 7 March 2021), the known miRNAs were quantified with the FeatureCounts program (https://www.biostars.org/p/259542/ Access date, 7 March 2021), whereas the prediction and quantification of novel miRNA were done with the miRDeep2 program (http://www.bioconductor.org/packages/release/bioc/vignettes/DESeq2/inst/doc/DESeq2.html Access date, 7 March 2021) [41]. For vsiRNA, the total clean reads were aligned with the different virus sequences of STV, CMV, and PepMV (GenBank accession numbers KJ174690.1, AB188234 and KJ018164).

### 2.5. Statistical Analysis

For plant symptoms, weight and height, and virus titer, the data were statistically analyzed using a mixed model PROC MIXED in the SAS software. Plant effect was included as a random effect, whereas time or inoculation was included as a fixed effect. Least Square Difference (LSD) was used for mean comparisons. The assumption of normal distribution of data was assessed using the normal probability plot of the residuals and the assumption of homoscedasticity using the Levene’s test. A 95% of confidence interval was considered in all cases. For miRNA differential expression analysis, a FDR adjusted *p*-value < 0.05 corresponding to a log Fold-change > 0.56 was considered to be statistically significant.

## 3. Results

### 3.1. Characterization of Field CMV and PepMV Isolates Used in this Work

The CMV and PepMV isolates showed nucleotide identities of 100% with the Japanese CM95 isolate (GenBank accession no. AB188236.1) in the 325 nt CP amplified region and with the European EU_CAHN8 isolate (GenBank accession no. JQ314457.1) in the 545 nt TGB3 amplified region, respectively. ELISA results analysis showed that PepMV and CMV isolates were no infected with other viruses, such as ToMV, TSWV, and PMoV: three replicates were used of each virus isolate and negative absorbance values were observed for ToMV (from 0.038 to 0.161), TSWV (from 0.047 to 0.075), and PMoV (from 0.039 to 0.059), whereas the positive control ranged from 0.903 to 2.076.

### 3.2. Effect of STV in Symptoms of Tomato Plants Mixed- Infected with PepMV and/or CMV

STV-infected tomato plants were single inoculated with PepMV or CMV, and with a combination of both viruses, to study the effect of STV in mixed infection. Leaf deformation and mosaic (severity scoring from 0 to 3, where 0 correspond to symptomless and 1, 2, and 3 to mild, moderate, and severe symptoms, respectively) were observed in infected tomato plants at different times (5, 10, 15, and 20 dpi) (Figure 1). Additionally, height and weight of tomato plants of different plant groups was taken at 20 dpi.

The leaf symptoms severity values (mean of plants symptoms on each group) in tomato plants infected with STV, CMV, and PepMV in single and/or mixed infections varied depending on the time and the virus combination (Figure 2). As expected, both STV single-infected and mock-inoculated plants remained symptomless for all the times [5,6]. At 5 dpi, only the STV + CMV co-infected plants showed mild symptoms (1.25), whereas no symptoms were observed in CMV-single infected plants. At 10 dpi, symptoms of STV + CMV co-infected plants were moderate (2.12), whereas those of STV + PepMV co-infected ones were mild (1.47). At this time, CMV and PepMV single-infected plants only showed mild symptoms (1.24 and 1.06, respectively). Regarding CMV + PepMV co-infection, these plants showed mild symptoms (1.00), whereas STV + CMV + PepMV triple-infected ones remained symptomless. At 15 dpi, the symptoms severity of STV + CMV co-infected plants decreased (from 2.12 to 1.55), whereas those of STV + PepMV double-infected ones increased (from 1.47 to 1.97). At this time, symptom severity of CMV single-infected plants (1.00) was lower than STV + CMV co-infected ones and higher in PepMV single-infected plants (2.63) than STV + PepMV co-infected ones. In contrast, the symptom severity of CMV + PepMV co- and STV + CMV + PepMV triple-infected plants was similar (1.43 and 1.37, respectively). Finally, at 20 dpi, symptom severity of STV + CMV and STV + PepMV co-infected plants decreased (from 1.55 to 1.26 and from 1.97 to 1.00, respectively) showing a remarkable difference respect to CMV single infection (2.62), but none with respect to PepMV-single infection (1.00). Regarding CMV + PepMV co-infected plants, symptom severity was slightly higher than STV + CMV + PepMV triple-infected ones (2.00 and 1.76, respectively).

Differences of height and weight (mean values) among groups of infected-tomato plants are shown in Figure 3. STV single-infected and mock-inoculated plants had similar height (64 and 65 cm, respectively). PepMV and CMV single-infected plants were significantly taller and smaller than the mock-inoculated ones (71.0 and 57.5 cm, respectively). The STV + PepMV co-infected plants (75.5 cm) were significantly taller than PepMV single-infected plants (71.0 cm), whereas height of STV + CMV co- and CMV single-infected plants was almost identical (57.7 and 57.5, respectively). The height of CMV + PepMV co-infected plants (64.0 cm) scored between STV + PepMV (75.5 cm) and STV + CMV (57.7 cm) co-infected plants, with significant differences with respect to both of them. Finally, STV + CMV + PepMV triple-infected plants were the smallest (50.0 cm), with significant differences with respect to the rest of virus-infected and mock-inoculated plants. With regard to the weight, mock-inoculated, STV, and CMV single-infected plants did not show significant differences (11.0, 11.9, and 12.3 g, respectively). PepMV single-infected plants reached the maximum value (32.7 g) of the assay, with significant differences with respect to the other plant groups, including STV + PepMV co-infection (21.7 g). The weight of STV + CMV co- and CMV single-infected plants was similar (13.2 and 12.3 g, respectively), whereas the weight of CMV + PepMV co-infected plants (15.5 g) scored between STV + PepMV and STV + CMV co-infected plants. Finally, the weight of STV + CMV + PepMV triple-infected plants (11.3 g) was significantly lower than that of CMV + PepMV co-infected ones.

### 3.3. Effect of STV in Virus Accumulation of Tomato Plants Mixed-Infected with PepMV and/or CMV

Virus accumulation was studied at 5, 10, 15, and 20 dpi by RT-qPCR using specific primers and TaqMan probes for STV, PepMV and CMV (Appendix A). The specificity assays of virus primer and probe sets showed no unspecific cross-amplifications. Figure 4 shows the mean values of virus accumulation. STV titer remained almost constant (2.38 × 10^4^ − 2.29 × 10^5^ virus RNA copies/ng total RNA) overtime for single-, co-, and triple-infections with CMV and PepMV, so the other viruses did not affect STV accumulation (Figure 4, Panel A). PepMV accumulation pattern was quite similar in single-, co-, and triple- infections (Figure 4, Panel B): this pattern consisted in a decrease from 5 dpi (2.70–5.28 × 10^6^ virus RNA copies/ng total RNA) to 15 dpi (1.39–2.43 × 10^5^ virus RNA copies/ng total RNA) and an increase at 20 dpi (1.01–5.20 × 10^6^ virus RNA copies/ng total RNA). CMV showed different accumulation patterns, depending on the virus combination (Figure 4, Panel C): the CMV concentration showed a low variation at 5 and 10 dpi (from 5.21 × 10^3^ to 1.64 × 10^4^ virus RNA copies/ng total RNA), but it increased strongly at 20 dpi (2.96 × 10^8^ virus RNA copies/ng total RNA) in CMV single infected plants. However, co-infection with STV produced a high increase of CMV at 10 and 15 dpi (8.30 × 10^5^ and 4.46 × 10^7^ virus RNA copies/ng total RNA, respectively), but, at 20 dpi, the CMV titer was similar in single- and STV + CMV co-infected plants (2.96 and 2.45 × 10^8^ virus RNA copies/ng total RNA, respectively). The pattern of CMV accumulation changed when co-infected with PepMV: the CMV titer decreased at 15 dpi (from 8.93 × 10^3^ to 1.05 × 10^2^ virus RNA copies/ng total RNA) and increased slightly it at 20 dpi (7.47 × 10^3^ virus RNA copies/ng total RNA). Differences of CMV accumulation between CMV + PepMV co-infected and CMV-single infected plants were significant at 10, 15, and 20 dpi. STV infection increased strongly CMV titer in STV + CMV + PepMV triple- infection at 10, 15, and 20 dpi (5.95 × 10^3^ 8.76 × 10^6^ and 4.16 × 10^7^ virus RNA copies/ng total RNA, respectively) to be similar to those of CMV single-infected plants. The differences of CMV accumulation were significant between STV + CMV + PepMV triple- and CMV + PepMV co-infection at 10, 15 and 20 dpi.

### 3.4. Effect of STV in siRNA Accumulation of Tomato Plants Mixed-Infected with PepMV and/or CMV

The accumulation of siRNAs was determined by high throughput small RNA sequencing from total nucleic acids obtained at 15 dpi, since the greatest effect of STV in CMV accumulation was found between 10 and 15 dpi. Additionally, at 15 dpi, a strong effect of STV in CMV + PepMV co-infection was observed. Three biological replicates were sequenced for each group of tomato infected plants and mock-inoculated plants were used as the controls. Biological replicates considered to be outlayer by PCA analysis were excluded from further analysis and the total reads were filtered to obtain the useful reads of about 21–24 nts (Appendix A). The percentages of useful reads with respect to the total ranged from 34% to 70%. The highest percentages of useful reads were found in the mock-inoculated and STV single-infected plants (59% and 70%, respectively), whereas, in the other virus-infected plants, they ranged from 34% to 48%. Expression profiling analysis of potential miRNAs performed with the OASIS 2 software showed a total of 78 siRNAs, which accumulated differentially in the plants that were infected with different virus combinations with respect to the control mock-inoculated plants (FDR < 0.05 and for log2FC > 0.56) (Table 1). Of those, 31 miRNAs were described previously in tomato and 47 corresponded to potential novel miRNAs described on other plant species, such as *Solanum tuberosum*, *Oryza sativa*, *Glycine max*, *Prunus persica*, or *Arabidopsis thaliana*. Three miRNAs with animal sequence homology, such as cow (*Bos taurus*) and mouse (*Mus musculus*), were also detected. It was found 5, 34, and 39 miRNAs with differential expression in STV, CMV, and PepMV single-infected plants, respectively. STV infection modified the number of miRNAs in STV + CMV and STV + PepMV co-infection with respect to CMV and PepMV single- infections (from 34 to 57 and from 39 to 37, respectively) (Appendix A). Slight changes in the number of miRNAs with differential expression were observed between the CMV + PepMV co-infected and the STV + CMV + PepMV triple-infected plants (from 24 to 25) (Appendix A). Finally, in CMV + PepMV co-infected plants, less miRNAs expressed differentially were found than in CMV and PepMV single-infected plants (from 24 to 34 or 39, respectively) (Appendix A). In addition to the change of the number of miRNAs expressed differentially, it was observed that STV infection significantly (FDR < 0.05 and fold-change was > 0.56) modified the accumulation of some miRNAs.

The potential functions of 53 out of these 78 miRNAs were determined by searching on the bibliography or by analysis with the online psRNAtarget software and they were mainly related with fundamental plant process, such as cellular biotic and abiotic stress, metabolism, or plant development. For example, it was reported that sly-miR9470-5p was related to hydric and salt stress, and Potato virus Y (PVY) infection as well [42,43]. This miRNA was upregulated in plants that were infected with all of the virus combinations with respect to the control mock-inoculated plants. Additionally, mtr-miR172c-5p that was previously related to salt stress [44] was up-regulated in CMV-single, and STV +CMV co-infected plants, whereas it was down-regulated in STV + CMV + PepMV triple-infected plats with respect to the control mock-inoculated. Finally, psRNAtarget analysis showed that mmu-miR-466i-5p that was upregulated in STV + CMV co-infected plants with respect to the control mock- inoculated ones targeted a gene encoding for a thylakoidal chloroplastic protein.

To obtain the vsiRNAs populations, the useful reads were aligned with the complete nucleotide sequence of STV, CMV, and PepMV (KJ174690.1, AB188234, and KJ018164). For each virus combination the percentage of vsiRNAs with respect the useful reads were calculated using the mean values of the biological replicates (Figure 5). It was detected few STV derived vsiRNAs in STV single-infected tomato plant (23.56 useful reads, which corresponded to 0.0003% of vsiRNAs), but they increased with the presence of other viruses in mixed-infections: STV + CMV and STV + PepMV co-infected plants (839.88 and 329.39 useful reads, which corresponded to 0.0106% and 0.0028% of vsiRNAS, respectively) and STV + CMV + PepMV triple-infected plants (1559.32 useful reads that corresponded to 0.0081% of vsiRNAs).

With regard to CMV derived vsiRNAs, high quantities were detected in CMV single- and STV + CMV co- infected plants (272,758.04 and 288,716.41 useful reads, which corresponded to 5.53% and 3.67% of vsiRNAs, respectively). However, in CMV + PepMV co-infected plants, CMV derived vsiRNAs were almost undetectable (7.82 useful reads, which corresponded to 0.000068% of vsiRNA), but they increased markedly with the STV presence in STV + CMV + PepMV triple-infected plants (330,588.95 useful reads which corresponded to 1.73% of vsiRNAs). Contrarily, PepMV derived vsiRNAs decreased in STV + PepMV co- and CMV + PepMV co-infected plants (8140.93 and 15,980.81 useful reads, which corresponded to 0.1446% and 0.1396% of vsiRNAs, respectively) with respect to PepMV-single infected ones (21,741.45 useful reads, which correspond to 0.2590% of vsiRNAs). However, in STV + CMV + PepMV triple infection (48,620.65 useful reads, which corresponded to 0.2551% of vsiRNAs), PepMV derived vsiRNAs increased with respect to PepMV single-infection.

The polarity of the vsiRNA plus (+) or minus (−) was also determined by aligning the useful reads with the positive and negative genomic virus strands (Table 2). For STV, in all virus combinations (STV single-, STV + CMV co-, STV + PepMV co-, and STV + CMV + PepMV triple-infected plants), more minus than plus vsiRNAs were detected (52.80–74.36% and 25.64–47.2%, respectively). For CMV, more minus than plus vsiRNAs were detected in CMV single-, STV + CMV co-, and STV + CMV + PepMV triple-infected tomato plants (67.39–72.37% and 27.35–32.61%, respectively) and less than CMV + PepMV co-infected ones (and 43.22 and 56.78%, respectively). For PepMV, similar amounts of plus and minus vsiRNAs were detected (50.35–53.39% and 47.25–50.41%, respectively).

Moreover, the distribution of the plus and minus vsiRNAs (average number of biological replicate) from each virus genome was determined by calculating the vsiRNAs frequency at each virus nucleotide position (Figure 6). STV and CMV vsiRNAs frequencies could not be represented in STV-single infected and CMV + PepMV co- infected tomato, since the amounts of vsiRNAs were so low. Both plus and minus vsiRNAs displayed a non-uniform distribution pattern along the virus genomes with hotspots (high accumulation of vsiRNAs) in specific genomic regions. These plus and minus vsiRNAs patterns were not symmetric for all viruses. For each virus, co-infection with the other virus did not produce remarkable variations of vsiRNAs patterns, but only changes in its accumulation level. Further estimations of vsiRNAs hotspots showed that minus STV vsiRNAs accumulated in the p42 (CP) coding region, which overlaps with the RdRp, meanwhile the plus vsiRNAs accumulated in the terminus part of the RdRp and the starting part of 3´ non-coding UTR. For CMV, plus and minus vsiRNAs accumulated more in the RNA3 (encoding for the MP and CP) than in the RNA2 (encoding for the RdRp and the 2b protein) and RNA1 (encoding for RdRP). In the RNA, mainly for the minus vsiRNAs, several hotspots were observed in the start and terminus parts of RdPp. In the RNA2 and 3, plus and minus hotspots were observed spread along genome, but in different positions, depending on the strand polarity. Some of the regions in the 2b, MP, and CP regions showed a high accumulation of vsiRNAs. Finally, for PepMV, plus and minus vsiRNAs hotspots localized along the virus genome but with some hotspots in the 5‘non-coding UTR, start part of the RdRp, TGB3, and CP regions.

## 4. Discussion

STV is a persistent virus that is widespread, and high incidences have been reported in some Spanish tomato production areas, such as Valencian community and Canary Islands. Despite that STV was associated to some disorders, such as a lack of fruit maturation and coloration alteration [3], recent studies suggest that STV does not produce symptoms in in tomato STV single-infected plants [5,6]. However, STV is frequently detected in tomato fields in combination with other viruses, but, to date, the effect of STV in mixed-infections on plant symptom development is unknown. In this work, the interaction of STV with two important acute viruses infecting tomato crops, such as CMV and PepMV, was studied. For this purpose, an assay with tomato plants in virus single-, co- and triple-infections was performed. As expected, STV-single infected plants did not show any symptoms, corroborating the results that were obtained in previous research works [5,6,7,45,46]. In this assay, the STV titer remained constant over time (5–20 dpi) in single-infections, as reported previously [5], and the same occurred in in co- and triple-infections with CMV and PepMV The steady titer of STV during the infection contrasts with the majority of acute viruses, whose concentration varies, depending on the infection state [38,39].

STV and CMV in co-infections established a synergistic interaction that produced the earlier apparition of leaf symptoms, increasing their severity, and increasing CMV titer in the first stages of infection. STV also produced an increase of plant symptoms in STV + PepMV co-infected plants, but it did not produce changes in PepMV titer. To our knowledge, this is the first report of a synergistic interaction between a persistent and two acute viruses. The best-known synergisms between acute viruses are those involving potyviruses. For example, infection with the potyvirus PVY and the potexvirus Potato virus X (PVX) increases the accumulation of PVX and the severity of symptoms [47,48]. It has been reported that potyviral VSR (HC-Pro) can suppress the defense mechanism that is based on the plant PTGS, favoring the replication and accumulation of the accompanying virus and enhancing the induced plant symptoms [24,49]. STV could codify for a VSR, but previous studies that were carried out in our lab showed that p42 had no VSR activity (unpublished data). Because STV only codifies for p42 and RpRd, further studies must be performed to confirm whether RdRp has VSR activity.

PepMV single-infected plants showed the maximum severity foliar symptoms (medium-severe) at 15 dpi and then decreases at 20 dpi. However, few changes of the viral titer were observed, with a slight decrease from 5 to 15 dpi and a recovery at 20 dpi. This accumulation pattern is not common in acute viruses, which normally increase the viral concentration at the beginning of the infection to reach a maximum that is followed by a stable or “plateau” stage, or sometimes with a slight decrease in the virus concentration [38,39]. Virus accumulation depends on many biotic and abiotic factors. For example, some Broad bean wilt virus 1 (BBWV-1) isolates showed abnormal accumulation patterns in pepper similar to that shown by PepMV in this study, whereas the same BBWV-1 isolates accumulated normally in tomato [38]. Furthermore, at 20 dpi, it was observed that PepMV infection induced an increase in the height and weight of the virus single-infected plants with respect to the control mock-inoculated ones. There are studies showing the beneficial effects of some acute viruses, as, for example, CMV that induces symptoms, but it is also able to increase the thermic resistance in beet *(Beta vulgaris)* infected plants [50]. The presence of STV in STV + PepMV co-infected plants increased their height respect to PepMV-single infected ones. However, in this plant group, the weight decreased as consequence of a stem slimming. It has been previously reported that the co-infection of PepMV and ToTV induces a slight increase in the height of infected tomato plants as compared to uninfected ones [51].

In CMV + PepMV co-infected plants, an antagonistic effect was observed with a decrease of CMV titer and symptoms were different to those that were induced by CMV or PepMV in single infections. Because CMV and PepMV are phylogenetically unrelated, this interaction cannot be explained as “cross protection” or “mutual exclusion”, which are produced between closely related viruses. To date, a few antagonistic interactions between phylogenetically distant viruses have been reported, but the mechanisms underlying these interactions have not been determined. For example, the simultaneous infection of Cucumber green mottle mosaic virus (CGMMV) and Tomato leaf curl New Delhi virus (ToLCNDV) in squash plants *(Cucurbita maxima)* led to a reduction in ToLCNDV titer, decreasing the virus-induced symptoms [52]. STV presence in plants STV + CMV + PepMV triple-infected plants suppressed the antagonistic effect between CMV and PepMV, restoring the CMV titer in CMV single-infected plants and modifying the symptom severity with respect to CMV + PepMV co-infection. To our knowledge, this is the first description of a virus modifying interaction being established between two other different viruses.

In this research work the effect of the interaction between STV, CMV, and PepMV on the populations of both plant miRNAs and viral vsiRNAs was also studied. The differential expression of 78 miRNAs was determined in tomato plants in single and mixed infection conditions with respect to the control mock-inoculated plants. Of all these miRNAs, 47 corresponded to novel miRNAs that were described for the first time in tomato. It was previously reported that plant infection by viruses, such as PVY and Papaya ringspot virus (PRSV), stimulated the synthesis of novel miRNAs [43,53]. miRNAs with differential expression that were found in this work were mainly involved in fundamental processes in the plant, such as development, metabolism, abiotic, and biotic stress. Thus, variations in accumulation of these miRNAs could lead to important changes in the plant. The number of miRNAs differentially expressed, and their level of accumulation, varied depending on the virus combination. Additionally, it was demonstrated that STV presence in the different groups of infected plants modified both the number and expression level of some miRNAs with respect to the CMV or PepMV single- and CMV + PepMV co-infections. Some examples of miRNAs with differential expression, depending on the virus combination, are: mtr-miR172c-5p was up-regulated in CMV-single, and STV +CMV co-infected plants, whereas it was down-regulated in STV + CMV + PepMV triple-infected plants with respect to the control mock-inoculated ones. This miRNA was previously related to salt stress [44]. sly-miR164b-3p had differential expression in STV + CMV co-infected plants, but not in CMV or STV single-infected ones (Appendix A). The miRNA was related to saline and hydric stress as well as fruit maturation in tomato plants [42,54]. miRNA stu-miR398a-5p had differential expression in STV + PepMV double-infected plants, but not in PepMV or STV single- infected ones (Supplementary material S4). This miRNA is related to tolerance to the virus infection [55]. Finally, mmu-miR-466i-5p was upregulated in STV + CMV co-infected plants with respect to the control mock-inoculated ones (Supplementary material Table S3). This miRNA can target a gene encoding for a thylakoidal chloroplastic protein. In the last years, reports of changes in the miRNA expression as consequence of plant virus infection have been increasing. For example, miR159/319 and miR172 expression is modified by ToLCNDV infection in tomato [56], or miR163, miR164, and miR167 expression is modified by ToMV infection in *A. thaliana* [57].

STV, CMV, and PepMV plus and minus vsiRNAs were identified. The results obtained in this work showed that the vsiRNAs proportion varied, depending on the virus (single infection) and the combination whit other viruses (multiple infection). The amount of vsiRNAs generated in STV single-infected plants was very small, but it increased markedly in plants that were co-infected with PepMV or CMV (STV + PepMV and STV + CMV co-infections). These variations in the production of vsiRNAS from STV were not related to viral accumulation, since the concentration of STV did not change with the presence of other viruses. This is in concordance with the results reported by other authors that showed low vsiRNAs concentrations in STV single-infected tomato plants, but that increased in combination with other viruses in mixed-infections [46,58,59]. It was observed that the presence of STV varied the proportion of CMV and PepMV vsiRNAs in STV + CMV and STV + PepMV co-infections with respect to the CMV and PepMV-single infections, and that these changes were not related with the viral accumulation. The interaction of CMV and PepMV also influenced the formation from vsiRNAs in both viruses in co-infections, mainly with a strong reduction of CMV vsiRNAs, which were practically not detected. In this case, the low titer of CMV vsiRNAs correlated with the low titer of CMV RNA due to the antagonistic effect of PepMV. Differences in the vsiRNAs accumulation may be relevant in the development of plant symptoms due them having the ability to mimic the miRNAs by sequence homology. For example, there is experimental evidence that some vsiRNAs of the Sugarcane mosaic virus (SCMV) and Rice stripe virus (RSV) target genes in corn and rice, respectively, altering their development [22]. Additionally, vsiRNAs are generated by CMV satellite RNA Y in *N. tabacum* and by Tomato yellow leaf curl virus (TYLCV) in tomato target genes of these plants [22,23,60].

The study of the frequency of STV, CMV, and PepMV derived vsiRNAs per nucleotide site, in the positive and negative strands of the viral genomes, showed that the distribution of plus and minus vsiRNAs was not uniform in these viruses. It detected regions of accumulation with peaks (hotspots) that were usually different for the plus and minus vsiRNAs. However, the distribution patterns of vsiRNAs for each virus were not influenced by the presence of the other viruses in mixed infection. Differences were only observed in the vsiRNAs accumulation level, which was correlated with the number of total useful reads. This agrees with the non-uniform patterns of STV and PepMV vsiRNAs frequencies found by other authors [59]. Additionally, it has been reported that co-infections of PRSV and PapMV did not alter the its frequency patterns of vsiRNAs accumulation with respect to simple infections [47].

## 5. Conclusions

To date, the role played by STV in the development of some plant symptoms, such as disorders in tomato fruit coloration and maturation, was controversial. Despite recent studies showing that STV did not induce any plant symptoms in single-infections, the reported symptoms could be induced by the interaction of STV with other viruses. Here, we studied the effect of STV in co- and triple-infections with the widespread Cucumber mosaic virus (CMV) and Pepino mosaic virus (PepMV). The results showed that the persistent STV is relevant from a phytopathological point of view, since STV can interact with these viruses: (i) establishing a synergism with CMV or PepMV in which STV increased CMV titer and CMV induced symptoms at early infection stages, whereas PepMV titer did not change in spite that PepMV induced symptoms exacerbated, (ii) suppressing the antagonism between CMV and PepMV, restoring the CMV titer, and modifying the plant symptom severity with respect to CMV + PepMV co-infection, and (iii) modifying the accumulation of both plant miRNAs and viral vsiRNAs with respect to PepMV and CMV in single- or co-infections. Most of these miRNAs are involved in essential plant process, Additionally, vsiRNAs could mimic the action of miRNAs targeting plant genes. Thus, it is important to establish control measures to avoid STV spread by preventing the commercialization of STV-infected seeds, since the virus is only horizontally transmitted.

## Figures and Tables

**Figure 1 microorganisms-09-00689-f001:**
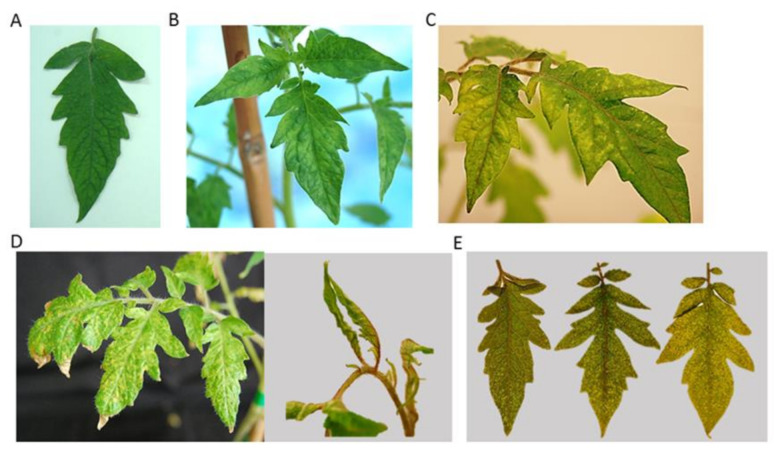
Mosaic and deformation leaf symptoms showed by tomato plants infected with Southern tomato, Pepino mosaic, and Cucumber mosaic viruses (STV, PepMV, and CMV, respectively) in single and mixed infections. Symptoms were considered as mild, moderate or severe (Panels **B**, **C**, and **D**, respectively). The right part of (Panel **D**) shows a strong leaf deformation in the plant shoots in tomato plants with severe symptoms. Panel **A** shows a symptomless leaf corresponding to a mock-inoculated tomato plant. (Panel **E**) shows three tomato leaves showing mild, moderate, or severe symptoms (from left to the right).

**Figure 2 microorganisms-09-00689-f002:**
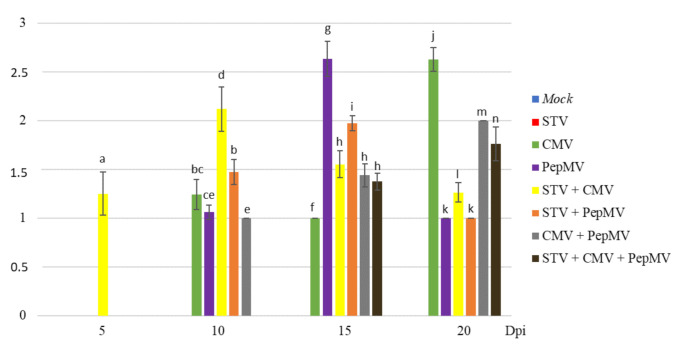
Graphic representation (mean values) of leaf symptoms severity (ordinate axis) of tomato plants infected with STV, PepMV, and CMV in single and mixed infections at 5, 10, 15, and 20 dpi (abscise axis). Leaf symptoms intensity was scored from 0 to 3, where 0 corresponds to symptomless, and 1, 2, and 3 to mild, moderate, and severe symptoms, respectively. Bars and letters up to the columns correspond to standard errors (from 0 to 0.23) and different plant groups (*p*-value ≤ 0.05), respectively.

**Figure 3 microorganisms-09-00689-f003:**
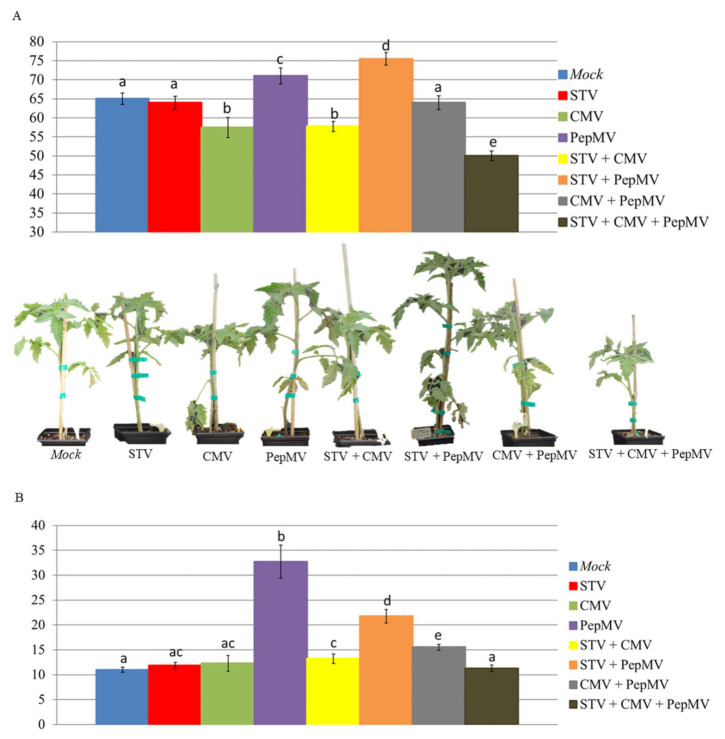
Graphical representation (mean values) of height (Panel **A**) and weight (Panel **B**) measured in cm and g, respectively, of tomato plants infected with STV, PepMV, and CMV in single and mixed infections at 20 dpi. Bars and letters up to the columns correspond to standard errors and different plant groups (*p*-value ≤ 0.05), respectively. At the bottom of (Panel **A**), we show the height of tomato plants infected with different virus combinations in comparison with mock-inoculated plants.

**Figure 4 microorganisms-09-00689-f004:**
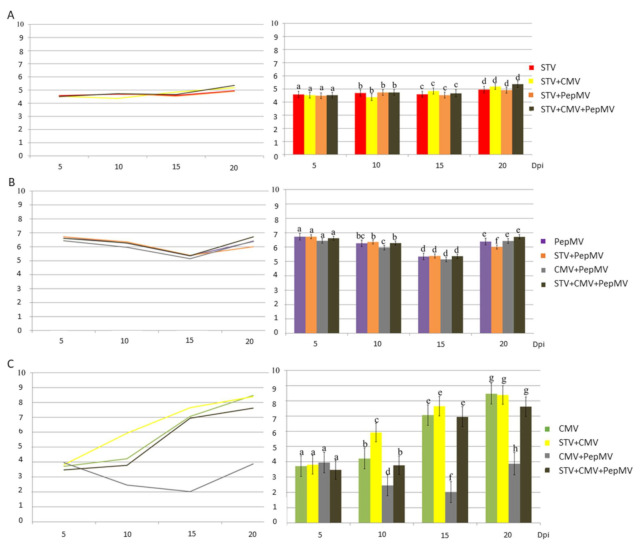
Virus accumulation (mean values) in tomato plants (ordinated axis) shown as log of concentration (no. RNA copies/ng of total RNA) of STV, PepMV, and CMV (Panel **A**, **B**, **C**, respectively) in single and mixed infections at 5, 10, 15, and 20 dpi (abscise axis). Bars and letters up to the columns correspond to standard errors and plant groups (in each dpi), respectively, showing differences (*p*-value ≤ 0.05). In each panel, virus accumulation is represented by columns (**right**) and in lineal representation (**left**).

**Figure 5 microorganisms-09-00689-f005:**
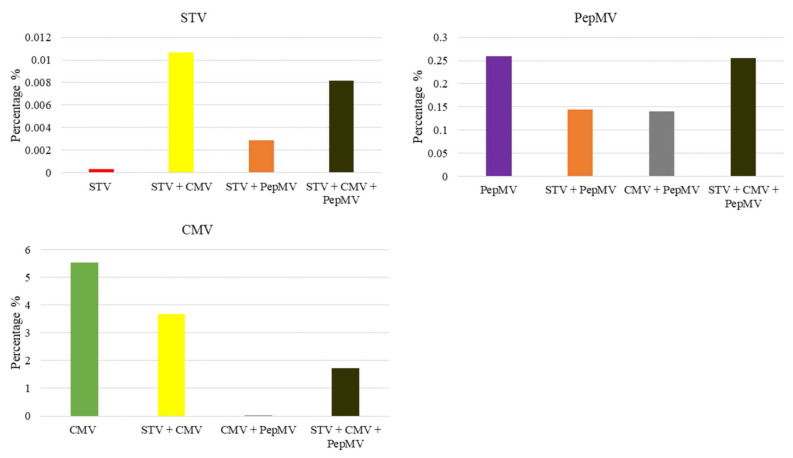
Graphic representation of the percentages (%) of vsiRNAs respect to the useful reads obtained by small RNA high throughput sequencing of the different STV, CMV, and PepMV virus combinations. The percentages of vsiRNAs were obtained from the useful reads mean values of the three biological replicates.

**Figure 6 microorganisms-09-00689-f006:**
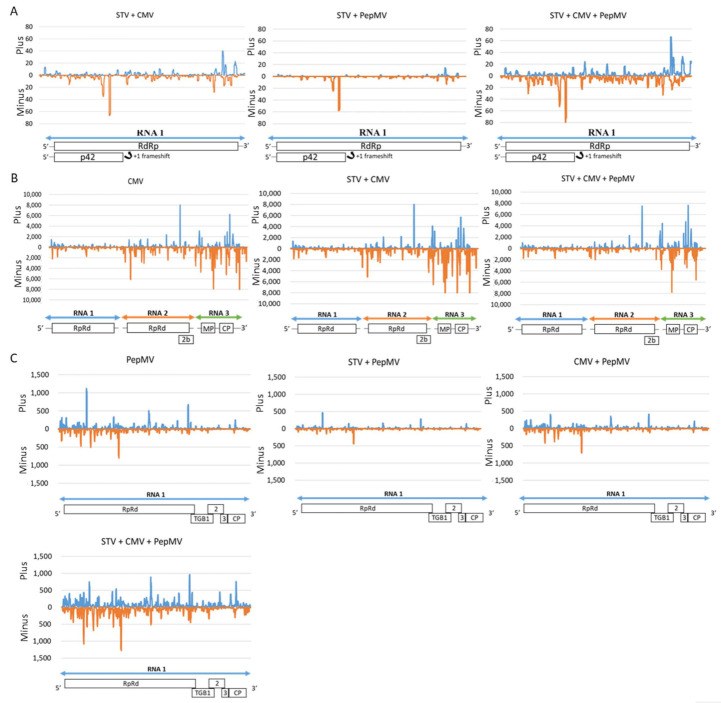
Graphical representation of plus (+) (blue) and minus (−) (orange) vsiRNAs frequencies along STV (Panel **A**), CMV (Panel **B**), and PepMV (Panel **C**) genomes for the different virus combinations. Virus genome organization is showed in the bottom of each graph. STV and CMV vsiRNAs frequencies were not represented, since amounts of vsiRNAs were so low in STV-single infected and CMV + PepMV double- infected tomato plants.

**Table 1 microorganisms-09-00689-t001:** Differential accumulation of miRNAs in tomato plants infected with different virus combinations with respect to the control mock-inoculated plants (FDR < 0.05 and log2FC > 0.56).

	miRNA with Differential Expression	
Sample	*Solanum lycopersicum*	Novel *Solanum lycopersicum*
STV vs. Mock-inoculated	1	4
CMV vs. Mock-inoculated	14	20
PepMV vs. Mock-inoculated	14	25
STV + CMV vs. Mock-inoculated	26	31
STV + PepMV vs. Mock-inoculated	15	22
CMV + PepMV vs. Mock-inoculated	11	13
STV + CMV + PepMV vs. Mock-inoculated	10	15

**Table 2 microorganisms-09-00689-t002:** siRNAs polarity (plus or minus) in tomato plants infected with de different STV, CMV, and PepMV virus combinations. The numbers for each virus combination correspond to the mean of the biological replicates. The percentages of plus (+) and minus (−) vsiRNAs polarity base on the useful reads are in brackets.

	STV		CMV		PepMV	
	Plus (+)	Minus (−)	Plus (+)	Minus (−)	Plus (+)	Minus (−)
STV	11.12	12.44	-	-	-	-
	(47.2%)	(52.80%)	-	-	-	-
CMV	-	-	88,928.90	183,829.14	-	-
	-	-	(32.61%)	(67.39%)	-	-
PepMV	-	-	-	-	11,469.85	10,271.60
	-	-	-	-	(52.75%)	(47.25%)
STV + CMV	332.56	507.32	78,955.78	209,760.63	-	-
	(39.6%)	(60.40%)	(27.35%)	(72.65%)	-	-
STV + PepMV	84.43	244.96	-	-	4346.79	3794.14
	(25.64%)	(74.36%)	-	-	(53.39%)	(46.61%)
CMV + PepMV	-	-	4.44	3.38	8047.01	7933.80
	-	-	(56.78%)	(43.22%)	(50.35%)	(49.65%)
STV + CMV + PepMV	626.67	932.65	90,349.20	240,239.75	24,113.38	24,507.27
	(40.19%)	(59.81%)	(27.33%)	(72.67%)	(49.59%)	(50.41%)

## Data Availability

All nucleotide sequences obtained in this study were uploaded to NCBI GenBank under accession numbers MT785769 and MT785770. Dataset obtained in the High-throughput small RNA sequencing was uploaded to the NCBI and published under the Bioproject PRJNA625104 and PRJNA574043.

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
