# Peer review of "Persistent Southern Tomato Virus (STV) Interacts with Cucumber Mosaic and/or Pepino Mosaic Virus in Mixed- Infections Modifying Plant Symptoms, Viral Titer and Small RNA Accumulation"

_microorganisms, 2021, doi:10.3390/microorganisms9040689_

Round 1
Reviewer 1 Report
Very interesting work. The paper highlights the complex interactions between viruses when exposed to multiple infections and the potential important role of persistent viruses in modulating these interactions and disease development. Paper is well written, but it requires minor changes in the English language. Below a few questions and minor comments about the manuscript that, in my view, will improve it.
line 67: “presence of RNA satellite molecules” affect CMV symptoms. were the host plants used in this study free of these satellite molecules?
line 118: why use conventional RT-PCR and qRT-PCR? why not qRT-PCR for all viruses?
line 119: what was the strain of PepMV used in the experiments? In introduction (line 54) authors states that “… symptom severity depends on several factors such as the virus strain, and the crop conditions”. Could the strain used in this work have an effect in the results obtained? Please discuss this.
Line 123: Is ELISA sensitive enough to detect these viruses if they are in low titre?
Line 162: in this section authors mention that RT-PCR was used for all viruses. why not sequence STV products too as you did for CMV and PepMV? Table S1 does not show information regarding the primers used for STV in conventional RT-PCR.
Line 168: did authors validate the qPCR primers for CMV? Please clarify.
Line 210: why 15dpi? Please add a sentence to justify using this datapoint for HTS analysis.
Line 223: could authors show the size distribution of sRNAs between the different treatments? any correlation?
Line 248: Please include the ELISA results. This is related to my previous question regarding line 123.
Line 264: are the leaf symptoms severity results statistically significant? Figure 2 does not show it. Please comment or re-do figure. What explains symptom severity going up and down?
Line 311: did authors measure fruit size/quality? It would be interesting to see what the difference between all different treatments would be.
Line 325: PepMV virus titre consisted in a decrease from 5 dpi to 15dpi and then increase at 20dpi. why? Discussion (lines 495-509) address this question but does not provide an explanation why PepMV is not behaving like an acute virus. could authors address this?
Line 344 (Figure 4): could authors clarify the letters in the graphs? is it related to each dpi or between dpi?
Line 363: big variation in the percentage of useful reads. Could you explain why?
Line 384: “expression level of some of them”. this sentence is vague. please edit this sentence.
Line 393-395: could authors provide a possible explanations on why “mtr-miR172c-5p” was up-regulated in CMV and STV+CMV and down regulated in triple infection?
Line 427: consider changing forward and reverse strands to plus (+) and minus (−) strands. also on table 3 and figure 6.
Line 483: in this study PepMV showed highest severity at 15dpi (figure 2) but virus titre was low at 15dpi (figure 4). could you explain these results?
Line 488: authors provide explanation for PVY. Could provide one for this study?
Author Response
Dear editor,
We are submitting the reviewed version of the manuscript ID: microorganisms-1154985 entitled “Persistent Southern tomato virus (STV) interacts with Cucumber Mosaic and/or Pepino mosaic virus in mixed- infections modifying plant symptoms, viral titer and small RNA accumulation”. All the referee´s issues have been attended. We hope that reviewed version of the manuscript is suitable for publication in Microorganisms Journal
Revisor 1
Very interesting work. The paper highlights the complex interactions between viruses when exposed to multiple infections and the potential important role of persistent viruses in modulating these interactions and disease development. Paper is well written, but it requires minor changes in the English language. Below a few questions and minor comments about the manuscript that, in my view, will improve it.
Line 67: “presence of RNA satellite molecules” affect CMV symptoms. were the host plants used in this study free of these satellite molecules?
Response: The CMV with CARNA 5 satellite isolates were important in the 90’s in Spain, but later the satellite disappeared in the virus populations. So, CARNA 5 is not present in Spain for many years ago and even the diagnosis laboratories don´t include satellite detection in the routine analysis. Also, no symptoms associated with the presence of the CARNA 5 satellite were observed in the CMV isolate used in this work (original field tomato plants nor tomato plants infected with this isolate in the mixed infection assay).
Line 118: why use conventional RT-PCR and qRT-PCR? why not qRT-PCR for all viruses?
Response: ToMV RT-qPCR detection was not available in our lab. However, we disposed of specific primers for the virus detection by conventional PCR.
Line 119: what was the strain of PepMV used in the experiments? In introduction (line 54) authors states that “… symptom severity depends on several factors such as the virus strain, and the crop conditions”. Could the strain used in this work have an effect in the results obtained? Please discuss this.
Response: Sequence analysis suggested that The PepMV isolate belonged to the European EU_strain (Results section “3.1 Characterization of Field CMV and PepMV Isolates Used in this Work” line 244). Also, the mild symptoms observed in the original field tomato plant, mainly leaf mosaic and leaf deformation without leaf or fruit necrosis, supported that. In our work, only mosaic and leaf deformation were observed in PepMV infected tomato plants. Of course, the use of other PepMV strains more severe could render different results and it is a very interesting issue for further studies.
Line 123: Is ELISA sensitive enough to detect these viruses if they are in low titre?
Response: It is known that molecular hybridization and PCR (conventional and qPCR) are more sensitive than ELISA. However, from my experience, ELISA is sensitive enough and robust for PMoV, ToMV, and TSWV detection from field samples. Also, collected field samples and tomato plants used in mixed infection assay didn´t show the typical necrosis symptoms induced by these viruses.
Line 162: in this section authors mention that RT-PCR was used for all viruses. why not sequence STV products too as you did for CMV and PepMV? Table S1 does not show information regarding the primers used for STV in conventional RT-PCR.
Response: This is a mistake that has been corrected in the new version of the manuscript (STV was removed, line 150). Conventional PCR was not used for STV detection (it was detected by RT-qPCR). Conventional PCR was only used in the case of STV for the elaboration of RNA transcripts (primers are detailed in Table S1). Also, STV nucleotide variability is so low, practically inexistent, and we cannot found different strains (Elvira-González et al. 2020). In the case of CMV and PepMV the scenario is quite different, and it was necessary to determine the strain which belonged. Finally, we have included the words “as described in section 2.2” (line 185) to clarify that RT-PCR conditions for STV amplification were the same that for CMV and PepMV amplification.
Line 168: did authors validate the qPCR primers for CMV? Please clarify.
Response: The specificity of primer and probe sets of all viruses was assessed and no unspecific cross-amplifications were observed. To explain that, we have added the sentence in the M&M section “The specificity of all virus primer and probe sets were assessed to avoid unspecific cross-amplifications (line 178) and “The specificity assays of virus primer and probe sets showed no unspecific cross-amplifications” (line 323) in the Results section.
Line 210: why 15dpi? Please add a sentence to justify using this datapoint for HTS analysis.
Response: We have chosen 15 dpi for HTS since the greatest effect of STV in CMV accumulation was found between 10 and 15 dpi. Also, at 15 dpi a strong effect of STV in the combination CMV and PepMV (CMV + PepMV double infection) were found. To justify this, we have added in the text (Results section, line 357) the sentence “since the greatest effect of STV in CMV accumulation was found between 10 and 15 dpi. Also, at 15 dpi, a strong effect of STV in CMV + PepMV co-infection was observed.”
Line 223: could authors show the size distribution of sRNAs between the different treatments? any correlation?
Response: Data provided by the “Supercomputational and Bioinnovation Center from the University of Málaga, Spain where HTS was performed not contained the distribution of vsiRNAs at 21, 22, 23, and 24 (individually) but only the mix of 21-24 for each genome. So, it was not possible to perform the frequency maps for each vsiRNA size.
Line 248: Please include the ELISA results. This is related to my previous question regarding line 123.
Response: ELISA tests were done several years ago when virus isolates were collected. We consider that negative ELISA data are no relevant in this research work but, following the referee advice, we have included the absorbance results in the text (Results section, line 248) “Three replicates were used of each virus isolate and negative absorbance values were observed for ToMV (from 0.038 to 0.061), TSWV (from 0.047 to 0.075) and PMoV (from 0.039 to 0.059) whereas the positive control ranged from 0.903 to 2.076”.
Line 264: are the leaf symptoms severity results statistically significant? Figure 2 does not show it. Please comment or re-do figure. What explains symptom severity going up and down?
Response: We performed Figure 2 with and without statistical analysis. Finally, we decided to include in the text the figure without statistical processing since symptoms are a qualitative and subjective issue. Also, we searched in the bibliography and find several works where symptoms data were no statistically analyzed (for example, Mixed infections of Pepino mosaic virus strains modulate the evolutionary dynamics of this emergent virus by Pedro Gomez et al. 2009). However, following the advice of both referees, Figure 2 with statistical analysis has been included in the text. We established a scale 0-3 of absolute values to put in data the symptom severity in leaf mosaic and deformation that we observed. The same two people (me and my assistant Laura Elvira) took always the symptoms at the same hour at the day to avoid light oscillations that difficult the symptom visualization. So, data were overall very homogeneous (in a few cases we put values of 1.5 or 2.5) and the errors very small being 0 in some cases. The inclusion of statistical data doesn´t change the results and the conclusion that we could extract of this experiment. We have added the error bars in Figure 2, and the following sentence (Results section, line 292) “Bars and letters up to the columns correspond to standard errors (from 0 to 0.23) and different plant groups (P-value ≤ 0.05), respectively.” Also, we have added the sentence “For plant symptoms, weight and height and virus titer…” In the text (M&M section, line 233).
Overall, we observed a logical or natural evolution in the leaf symptomatology of the different virus combinations: No symptoms were observed for STV and for PepMV, the symptomatology increases at 15 dpi, and later decreases as some acute virus. However, for CMV the symptomatology increases at 10 dpi, later decreases at 15 dpi and finally increases again at 20 dpi. But if we look at the graphic, the difference between 10 and 15 dpi is very small, approximately 0.2. We consider this difference very small in a subjective issue as the symptom observation.
Line 311: did authors measure fruit size/quality? It would be interesting to see what the difference between all different treatments would be.
Response: No, we did not measure the fruit or the quality of this fruit in this work. Of course, it is an interesting approach for further studies.
Line 325: PepMV virus titre consisted in a decrease from 5 dpi to 15dpi and then increase at 20dpi. why? Discussion (lines 495-509) address this question but does not provide an explanation why PepMV is not behaving like an acute virus. could authors address this?
Response: PepMV doesn´t follow a normal pattern of acute virus accumulation. However, sometimes, some acute viruses can show abnormal accumulation patterns. For example, several isolates of BBWV-1 showed in pepper host similar patterns to showed by PepMV (Carpino et al. 2018). So, virus accumulation depends on many factors such as host, virus isolate, and plant grow conditions.
We have included the sentence “Virus accumulation depends on many biotic and abiotic factors. For example, some BBWV-1 isolates showed abnormal accumulation patterns in pepper similar to that showed by PepMV in this study, whereas the same BBWV-1 isolates accumulated normally in tomato.” in the text (discussion section, line 503).
Line 344 (Figure 4): could authors clarify the letters in the graphs? is it related to each dpi or between dpi?
Response: Letters correspond to different plant groups (P-value ≤ 0.05) in each dpi and not between different dpi. The words “(in each dpi)” have been included after “plant groups” (Results Section, Line 351).
Line 363: big variation in the percentage of useful reads. Could you explain why?
Response: High throughput Small RNA sequencing was conducted in the Supercomputational and Bioinnovation Center from University of Málaga, Spain. The differences that we observed in the proportion of total and useful reads had a methodological origin. However, all samples reached the read range (quality standard) to detect the maximum of miRNA with differential expression. So, the differences in the proportion of total and useful reads observed did not disturb the estimation of miRNAs with differential expression.
Line 384: “expression level of some of them”. This sentence is vague. Please edit this sentence.
Response: We have changed in the text (Results section, line 384) the sentence “expression level of some of them” by “the accumulation of some miRNAs.”
Line 393-395: could authors provide a possible explanations on why “mtr-miR172c-5p” was up-regulated in CMV and STV+CMV and down regulated in triple infection?
Response: This is an example of how the different viral combinations can alter the miRNAs populations. In this case, the PepMV presence modifies the expression of mtr-miR172c-5p with respect to CMV and STV+CMV combination.
Line 427: consider changing forward and reverse strands to plus (+) and minus (−) strands. also on table 3 and figure 6.
Response: It was done
Line 483: in this study PepMV showed highest severity at 15dpi (figure 2) but virus titre was low at 15dpi (figure 4). could you explain these results?
Response: From a pathological point of view, the correlation between viral accumulation and symptom severity is sometimes no direct. Symptom development is the consequence of a complex pathway involving structural and ultra-structural changes in the plant tissues. Also, plant tissues showing severe symptoms sometimes are no good for virus quantitation since they are damaged and the virus replication conditions are no optimal. The prior days to symptom manifestation are usually the best for virus quantitation.
Line 488: authors provide explanation for PVY. Could provide one for this study?
Response: We can hypothesize that STV could codify for a VSR equivalent to HC-Pro in PVY. However, a previous study carried out in our laboratory suggested that p42 had not suppressor activity. Then, further studies with RdRp could be performed.
We have added in the text (Discussion section, line 493) the sentence “STV could codifies for a VSR but previous studies carried out in our lab showed that p42 had not VSR activity (unpublished data). As STV only codify for p42 and RpRd, further studies must be performed to confirm if RdRp has VSR activity·.
References:
Carpino, C.; Elvira-González, L.; Rubio, L.; Peri, E.; Davino, S.; Galipienso, L. A comparative study of viral infectivity, accumulation and symptoms induced by Broad bean wilt virus 1 isolates. J. Plant Pathol. 2019, 101, 275–281.
Elvira-González, L.; Medina, V.; Rubio, L., Galipienso, L. The persistent southern tomato virus modifies miRNA expresión without inducing symptoms and cell ultra-structural changes. Eur. J. Plant Pathol. 2020, 156, 615–622
Gomez, P.; Sempere, R. N.; Elena, S. F.; Aranda, M. A. Mixed infections of Pepino mosaic virus strains modulate the evolutionary dynamics of this emergent virus. J Virol. 2009, 83, 12378-12387
Reviewer 2 Report
Dear Authors,
I have an honor to review manuscript entiled “Persistent Southern tomato virus (STV) interacts with Cucumber mosaic and/or Pepino mosaic virus in mixed- infections modifying plant symptoms, viral titer and small RNA accumulation” submitted to Microorganisms MDPI Journal.
Authors presented interesting investigation focused on the effect of STV in co- and triple-infections with Cucumber mosaic virus (CMV) and Pepino mosaic virus (PepMV). Authors analysed new findings coming from studies concentrated on plant symptoms, virus RNA accumulation and miRNA and vsiRNA accumulation assessed in single, double and triple infections.
I have a question based on what kind of tools or what kind of analysis authors scoring plant symptoms ? what kind of factor or estimation decided that symptoms were postulated as a 1 or2 score in some kind of scale ? Moreover, [Figure2] In what kind of analysis mean values of symptoms were crated? where are in plot information from single infection?, Authors postulated no symptoms coming from single infection of STV [figure2], despite of that has influenced on co-infected two others viruses- How to explain this statement?
Primers to RT-qPCR were designed to capsid proteins of CMV and STV viruses, why in PepMV one of movement protein TGB2 was chosen instead of CP ?
Please, provide better resolution of symptoms photography in figure1;especially A,B,E
Figure 4 -the same situation – please, provide better resolution;
In my opinion table 1 could be translocate to supplementary files;
Please, delineate the future prospects to findings coming from the results and mixed infection tendency.
Author Response
Dear editor,
We are submitting the reviewed version of the manuscript ID: microorganisms-1154985 entitled “Persistent Southern tomato virus (STV) interacts with Cucumber Mosaic and/or Pepino mosaic virus in mixed- infections modifying plant symptoms, viral titer and small RNA accumulation”. All the referee´s issues have been attended. We hope that reviewed version of the manuscript is suitable for publication in Microorganisms Journal
Revisor 2
Dear Authors,
I have an honor to review manuscript entiled “Persistent Southern tomato virus (STV) interacts with Cucumber mosaic and/or Pepino mosaic virus in mixed- infections modifying plant symptoms, viral titer and small RNA accumulation” submitted to Microorganisms MDPI Journal.
Authors presented interesting investigation focused on the effect of STV in co- and triple-infections with Cucumber mosaic virus (CMV) and Pepino mosaic virus (PepMV). Authors analysed new findings coming from studies concentrated on plant symptoms, virus RNA accumulation and miRNA and vsiRNA accumulation assessed in single, double and triple infections.
I have a question based on what kind of tools or what kind of analysis authors scoring plant symptoms? what kind of factor or estimation decided that symptoms were postulated as a 1 or2 score in some kind of scale?
Response: We established a scale from 0 to 3 of absolute values to put in data the symptom severity (leaf mosaic and deformation) that we observed. Being 0 absence of symptoms, and from 1 to 3 the mild, moderate and severe symptoms. Also, other authors establish a scale trying to put in data a qualitative value and a subjective observation. We were very strict in the symptom observation: The same two people (me and my assistant Laura Elvira) always took the symptoms at the same hour at the day to avoid light oscillations that difficult the symptom visualization. So, data were very homogeneous although we recognize that symptom observation can be subjective. Following the advice of the other referee, we have included statistical analysis in Fig 2. Some virus combinations have standard error values of 0 since data were sometimes very homogenous.
Moreover, [Figure2] In what kind of analysis mean values of symptoms were crated? where are in plot information from single infection?
Response: See above
Single infections were also represented in Fig 2 (see Right part of the figure, with different colors). Single infected STV plants did not show any symptoms as mock-inoculated ones (Severity score of 0).
Authors postulated no symptoms coming from single infection of STV [figure2], despite of that has influenced on co-infected two others viruses- How to explain this statement?
Response: In this experiments, STV alone did not produce plant symptoms as previously reported (Puchades et al. 2017; Elvira-González et al. 2018; Elvira-González et al. 2019). However, STV seems to interact with CMV and PepMV modifying the symptoms induced by these viruses. Also, CMV and PepMV interact showing an antagonism that is broken by STV. These results support the importance of mixed infections in plant disease development.
Primers to RT-qPCR were designed to capsid proteins of CMV and STV viruses, why in PepMV one of movement protein TGB2 was chosen instead of CP?
Response: The primer and probe set used for PepMV detection and quantification by RT-qPCR in this work was previously developed and published (Ling et al. 2007) and allowed us to detect all kinds of PepMV strains. Also, this primer and probe set worked very well, so we didn´t need to develop additional primers and probes for PepMV detection and quantitation.
Please, provide better resolution of symptoms photography in figure1; especially A,B,E. Figure 4 -the same situation – please, provide better resolution.
Response The resolution of Fig 1 was improved. It was included in the text a Fig with more resolution than that of the first version of the manuscript. We did the same with Fig 4.
In my opinion table 1 could be translocate to supplementary files.
Response: Table 1 was trans-located to supplementary files.
Please, delineate the future prospects to findings coming from the results and mixed infection tendency.
Response: In my opinion, the results of this research work open new ways for further studies. Mixed infections are very frequent in the field, so we consider interesting the study of STV interaction with other tomato damaging viruses in addition to CMV and PepMV. Also, we need to assess the real impact of STV in tomato crops from a pathological point of view by studying in field surveys the incidence of STV in combination with other acute viruses. Studies to evaluate how the change in the accumulation of some miRNAs could affect the tomato plant are relevant to explain the effect of virus interaction in disease development.
.
References:
Puchades, A. V; Carpino, C.; Alfaro‐Fernandez, A.; Font‐San‐Ambrosio, M.I.; Davino, S.; Guerri, J.; Rubio, L.; Galipienso, L. Detection of Southern tomato virus by molecular hybridisation. Ann. Appl. Biol. 2017, 171, 172–178.
Elvira-González, L.; Carpino, C.; Alfaro-Fernández, A.; Font-San Ambrosio, M.I.; Peiró, R.; Rubio, L.; Galipienso, L. A sensitive real-time RT-PCR reveals a high incidence of Southern tomato virus (STV) in Spanish tomato crops. Spanish J. Agric. Res. 2018, 16, 1008.
Elvira-González, L.; Medina, V.; Rubio, L.; Galipienso, L. The persistent Southern tomato virus modifies miRNA expression without inducing symptoms and cell ultra-structural changes. Eur. J. Plant Pathol. 2019.
Ling, K.-S.; Wechter, W.P.; Jordan, R. Development of a one-step immunocapture real-time TaqMan RT-PCR assay for the broad spectrum detection of Pepino mosaic virus. J. Virol. Methods 2007, 144, 65–72.